# Validity and Reliability of Kinect v2 for Quantifying Upper Body Kinematics during Seated Reaching

**DOI:** 10.3390/s22072735

**Published:** 2022-04-02

**Authors:** Germain Faity, Denis Mottet, Jérôme Froger

**Affiliations:** 1Euromov Digital Health in Motion, Univ Montpellier, IMT Mines Ales, 34090 Montpellier, France; germain.faity@umontpellier.fr; 2Euromov Digital Health in Motion, Univ Montpellier, IMT Mines Ales, CHU de Nîmes, 30240 Le Grau du Roi, France; jerome.froger@chu-nimes.fr

**Keywords:** Kinect, reaching, stroke, rehabilitation, upper limb, PANU, kinematics, markerless, mocap, motion capture

## Abstract

Kinematic analysis of the upper limbs is a good way to assess and monitor recovery in individuals with stroke, but it remains little used in clinical routine due to its low feasibility. The aim of this study is to assess the validity and reliability of the Kinect v2 for the analysis of upper limb reaching kinematics. Twenty-six healthy participants performed seated hand-reaching tasks while holding a dumbbell to induce behaviour similar to that of stroke survivors. With the Kinect v2 and with the VICON, 3D upper limb and trunk motions were simultaneously recorded. The Kinect assesses trunk compensations, hand range of motion, movement time and mean velocity with a moderate to excellent reliability. In contrast, elbow and shoulder range of motion, time to peak velocity and path length ratio have a poor to moderate reliability. Finally, instantaneous hand and elbow tracking are not precise enough to reliably assess the number of velocity peaks and the peak hand velocity. Thanks to its ease of use and markerless properties, the Kinect can be used in clinical routine for semi-automated quantitative diagnostics guiding individualised rehabilitation of the upper limb. However, engineers and therapists must bear in mind the tracking limitations of the Kinect.

## 1. Introduction

Stroke results in major movement deficits, especially in the upper limbs. Stroke survivors have reduced range of motion in the upper limbs and impaired elbow-shoulder coordination, partially compensated by increased trunk involvement in upper limb movements [1]. Their paretic upper extremity suffers from a reduced spatial and temporal efficiency, including decreased speed, increased number of velocity peaks and a longer path to reach a target [2]. To provide the best possible rehabilitation, therapists regularly assess the motor performance of individuals with stroke. However, the clinical scales used by therapists suffer from several limitations. First, clinical scales have a subjective scoring system which limits the reliability of ratings between therapists over time [3]. Second, clinical scales are too often insensitive to changes. For example, in most of its items, the Fugl–Meyer Assessment only supports 3 rating levels (0, 1 or 2), although it is considered one of the strongest clinical scales [4]. Third, most clinical scales do not sufficiently account for compensations that may occur in stroke movement [5,6], which cancels out the differences between a true recovery and a compensation pattern [7].

To go beyond these limitations, scientists use motion capture to quantify the motor deficits. Indeed, upper limb and trunk kinematics are more sensitive to changes than clinical scales [8,9], and can even predict motor outcomes over several months [10,11]. Moreover, motion capture makes compensation assessment easy and more objective [5,12,13]. Despite these advantages, kinematic assessment of the upper limbs remains little used in clinical practice because of its poor feasibility. Indeed, motion capture systems are expensive and require a large volume to perform the movements. In addition, patients have to be suited up with markers placed with accuracy, which is precarious and takes time in the clinical context. Finally, these technologies require a high technical level to extract valuable variables from raw data.

Since its release in 2013, the Kinect v2 (also known as Kinect One or Kinect for Xbox One, Microsoft, Redmond, WA, USA) has been widely used for rehabilitation purposes and has largely contributed to the rise of virtual reality in rehabilitation trials. Virtual reality with Kinect may be beneficial in improving upper limb function and activities of daily living when used as an adjunct to usual care (to increase overall therapy time) [14,15]. However, although the markerless and ease of use properties of the Kinect v2 facilitate its use in clinical routine, and its value for gait analysis has been documented [16], the validity of the Kinect for assessing upper limb kinematics after stroke remains to be tested.

Previous works on healthy subjects show that the Kinect v2 has an average accuracy of 10–15 mm but can generate distance errors up to 80 mm [17,18]. In addition, the Kinect would detect range of motion (ROM) with 1 to 10° error [19,20,21,22] but this result should be taken carefully given the variability between the studies. Indeed, some authors argue that the Kinect v2 has excellent reliability, especially for flexion of the elbow and shoulder [23], but others conclude that the kinematics obtained by the Kinect are unreliable [24] and that the use of IMUs should be preferred for motor assessment [25]. Most of these works were conducted either in static [17,19] or in low-functional situations [18,20]. For studies assessing upper limb movement, only the ranges of motion of the arm and trunk were assessed [21,23,25]. Finally, when measured, the other kinematic variables were not compared to ground-truth values [22]. Therefore, there is still a need for a validation of the Kinect tool for the study of upper limb kinematics.

Previous works on people with stroke show that, with the Kinect v2, hand and trunk range of motion are valid and reliable [26] and a combination of hand efficiency, hand smoothness and shoulder adduction can distinguish the reaching performance between healthy control, the less-affected side and the more-affected side of patients with stroke [27,28,29]. Yet, measures of temporal and spatial efficiency, though widely used in virtual reality rehabilitation [30], have not yet been validated.

The goal of our study is to investigate to what extent the Kinect v2 is valid and reliable for the kinematic assessment of reaching movements for application in post-stroke rehabilitation. To do so, we simultaneously recorded reaching movements with the Kinect v2 and with the Vicon motion capture system, which is considered the gold standard. We then assessed the validity and reliability of the Kinect v2 for key variables in upper limb kinematic assessment after stroke. We hypothesized that the Kinect v2 will provide the same information as the Vicon system.

Because it was not possible to ask patients to come to the laboratory, we tested the reliability of the Kinect v2 with a model of stroke behaviour, that is, with healthy subjects for whom post-stroke-like reaching behaviour was induced [31]. Specifically, we asked healthy participants to perform a series of reaching movements with their hand loaded to 75% of their maximum voluntary antigravity torque. In this condition, healthy participants spontaneously develop compensations similar to those observed in most stroke patients, including trunk flexion and rotation, reduced shoulder abduction, reduced movement velocity and increased path length ratio [31].

## 2. Materials and Methods

### 2.1. Participants

Twenty-six healthy participants (12 males, age 21 ± 3 years, 3 left-handed, height 1.73 ± 0.09 m, weight 66.92 ± 9.29 kg) took part in this study.

The inclusion criteria was to be aged between 18 and 40 years. Participants were excluded if they had shoulder pain or other problems that could affect their movement. This study was performed in accordance with the 1964 Declaration of Helsinki. The local ethics committee approved the study (IRB-EM 1901C).

### 2.2. Experimental Protocol

Participants had to reach a target with the side of their thumb nail. The target was a table tennis ball fixed in front of the participant at a height of 0.80 m, just within the anatomical reaching distance for the hand. The starting position was seated, feet on the ground, back in contact with the chair and forearms on the armrests.

In order to compute the Proximal Arm Non-Use (PANU) score [5], participants completed 5 reaches both in the spontaneous trunk use condition and in the restrained trunk use condition. In the spontaneous trunk use condition, participants had to reach the target at a natural pace, wait 1 s and return to the starting position. In the restrained trunk use condition, participants had to reach the target while minimizing trunk movement. During the restrained trunk use condition, the experimenter lightly touched the participants’ shoulders to remind them to minimize trunk movement. We did not use a belt to restrain the trunk in order to leave the participant free to use the trunk if necessary, and thus avoid task failure. The PANU score was then computed as the difference of shoulder-elbow use between the spontaneous condition and the restrained trunk condition [5].

The assessed hand was chosen pseudo-randomly (12 left, 14 right) so that half of the participants performed the task with their dominant hand, and the other half with their non-dominant hand. The weight of the arm including the dumbbell was set to 75.0% (±5.5%) of the maximum antigravity force (MAF) in the posture with the hand at the target.

### 2.3. Experimental Setup

The movements of the participants were recorded by both a Vicon motion capture system and a Kinect v2. The data obtained by the Kinect were then compared to the data obtained by the Vicon, the latter being considered the ground truth.

#### 2.3.1. Vicon Sensor

The Vicon system (Oxford Metrics, Oxford, UK) is a marker-based optoelectronic motion capture tool that is widely used for kinematic measurements [32]. Indeed, with a similar setting to ours, the error of the Vicon is 0.15 mm ± 0.025 mm in static and remains less than 2 mm in dynamic [33]. In this study, we used a 6-camera rectangle Vicon system with a sampling frequency of 100 Hz. Vicon time series were recorded using “Vicon Nexus 2” software (Oxford Metrics, Oxford, UK).

#### 2.3.2. Kinect Sensor

The Kinect v2 (Microsoft, Redmond, WA, USA) is a markerless motion capture tool combining 3 sensors (a RGB colour camera, a depth sensor and an infrared sensor) to provide the 3D position of 25 landmarks on a skeleton with a sampling rate of 30 Hz [26]. The Kinect was connected to a PC running the “MaCoKi” software (NaturalPad, Montpellier, France) developed from the Kinect SDK (v2.0_1409, Microsoft, Redmond, WA, USA) to record the position time-series of the hands and trunk. As recommended in previous studies, we placed the Kinect in front of the participant, at a distance of 1.50 m, a height of 1.40 m and with no direct sunlight to minimise errors [20,34,35,36] (Figure 1).

#### 2.3.3. Position of Landmarks for the Vicon and for the Kinect

In order to compare the Kinect data to the Vicon data, we placed the Vicon markers as close as possible to the joint centers located by the Kinect. Thus, we placed markers at the manubrium (spine-shoulder for Kinect), and for each body side on the first metacarpal (wrist for Kinect), the lateral epicondyle of the humerus (elbow for Kinect), the acromion process (shoulder for Kinect), and the anteriosuperior iliac spine. For each side, we corrected the anteriosuperior iliac spine marker position before data analysis to best match the anatomical center of the hip joints (hips for Kinect). In order to facilitate the reading, the spine shoulder marker is renamed “trunk marker” in the text.

### 2.4. Data Processing

Data processing was performed with SciLab 6.0.2. The Kinect being positioned obliquely to the ground to minimise visual occlusions of the elbow, we realigned the Kinect axes to match the Vicon axes. This realignment was performed using a solid transformation of the Kinect data based on the “Least-Squares Fitting” method [37] implemented in a Matlab function by Nghia Ho [38].

Because the Kinect errors produce high frequency noise (Figure 2), all position time series were low pass filtered at 2.5 Hz with a dual pass second order Butterworth filter. We chose a cut-off frequency of 2.5 Hz because the raw data showed that the frequency band 0–2.5 Hz contains at least 95% of the spectral density of the time series data.

We first calculated the start and end of each reaching movement in the one-dimensional task space [31]. Because the goal of a reaching task is to bring the hand to the target, that is, to reduce the hand-to-target distance, what is important for task success is the hand-to-target Euclidean distance. The hand-to-target Euclidean distance summarises the 3D effector space into a 1D task space (where movement matters) leaving aside a 2D null space (where movement does not impact task success). We fixed the beginning of the movement (*t*0) when the Euclidean velocity of the hand in the task space became positive and remained positive until the maximum velocity. The end of the movement (*tfinal*) was when the Euclidean distance to the target reached its minimum.

Angles presented in this study were calculated as the difference between the anatomical angle at *tfinal* and the anatomical angle at *t*0, as described in the Equation (1).
(1)$$retained\;angle=anatomical\;angle_{tfinal}-anatomical\;angle_{t0}$$

### 2.5. Statistical Analysis

We assessed the degree of reliability between the Kinect and the Vicon variables using intraclass coefficient correlation (ICC), coefficient of determination (r^2^), Root Mean Square Error (*RMSE*) and normalised Root Mean Square Error (*NRMSE*). We computed *RMSE* using Vicon values as ground truth, such as:(2)$$RMSE=\sqrt{\frac{\sum_{n=1}^{N}{\left(X_{vicon}-X_{kinect}\right)}^{2}}{N}}$$
where $X_{vicon}$ is the value measured by the Vicon, $X_{kinect}$ is the value measured by the Kinect and *N* is the number of observations. To facilitate the comparison between variables, we divided *RMSE* by the range of the variable, such as:(3)$$NRMSE=100\times \frac{RMSE}{\max\left(X_{vicon}\right)-\min\left(X_{vicon}\right)}$$

We complemented these measures with Bland and Altman plots to evaluate the validity of the Kinect through the difference in means and to estimate an agreement interval through the 95% limits of agreement [39]. To compare validity across variables, a relative systematic error was calculated as:(4)$$error_{relative}=\frac{mean_{vicon}-mean_{kinect}}{mean_{vicon}}$$

ICC estimates were calculated using R (version 3.6.1) based on a single-rating, consistency, one-way random-effect model. As stated by Koo & Li, “values less than 0.5 are indicative of poor reliability, values between 0.5 and 0.75 indicate moderate reliability, values between 0.75 and 0.9 indicate good reliability, and values greater than 0.90 indicate excellent reliability” [40]. We used the same limits for the *error_relative_*. The level of significance for all tests was set at *p* < 0.05. All coefficients of determination r^2^ were found to be statistically significant.

## 3. Results

The results are reported in Table 1.

Angles

**Figure 3 sensors-22-02735-f003:**
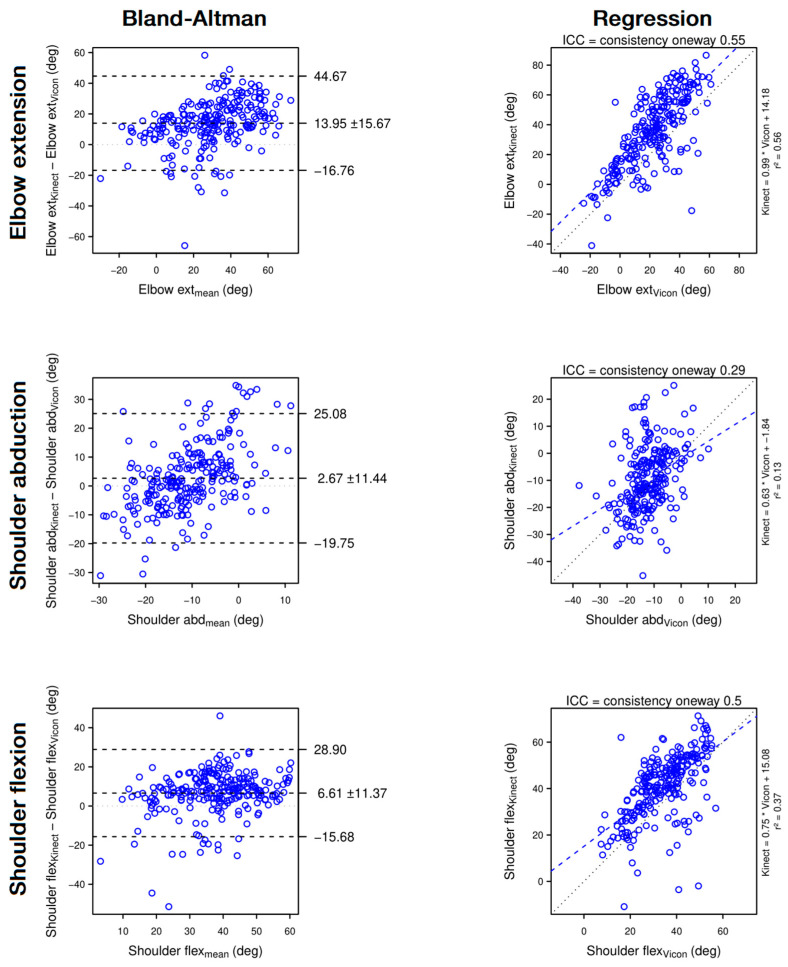
Bland–Altman plots (**left panels**) and regression plots (**right panels**) of elbow extension (1st row), shoulder abduction (2nd row) and shoulder flexion (3rd row). The Bland–Altman plots show that the Kinect strongly overestimates elbow extension, and slightly overestimates shoulder abduction and shoulder flexion. The regression plots show that, when assessed with the Kinect, elbow extension and shoulder flexion are moderately reliable and shoulder abduction is poorly reliable.

**Figure 4 sensors-22-02735-f004:**
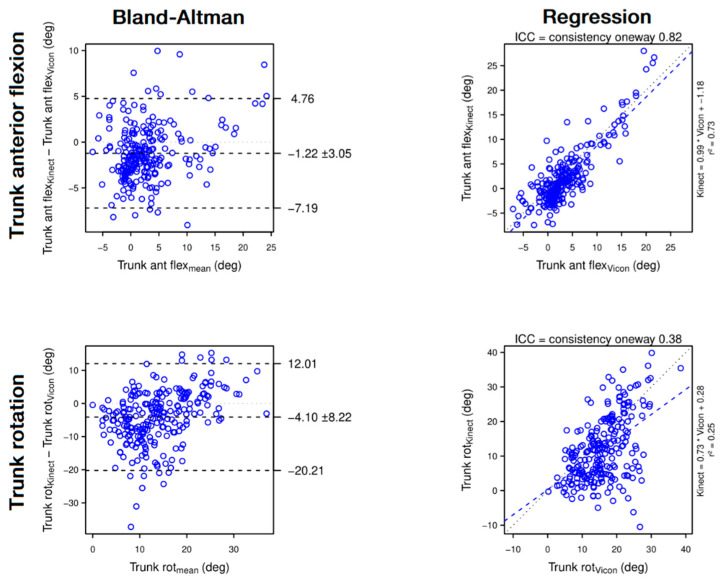
Bland–Altman plots (**left panels**) and regression plots (**right panels**) of trunk anterior flexion (1st row) and trunk rotation (2nd row). The Bland–Altman plots show that the Kinect moderately underestimates trunk anterior flexion and trunk rotation. The regression plots show that, when assessed with the Kinect, trunk anterior flexion is reliable, but trunk rotation is poorly reliable.

Efficiency, Planning and Smoothness

**Figure 5 sensors-22-02735-f005:**
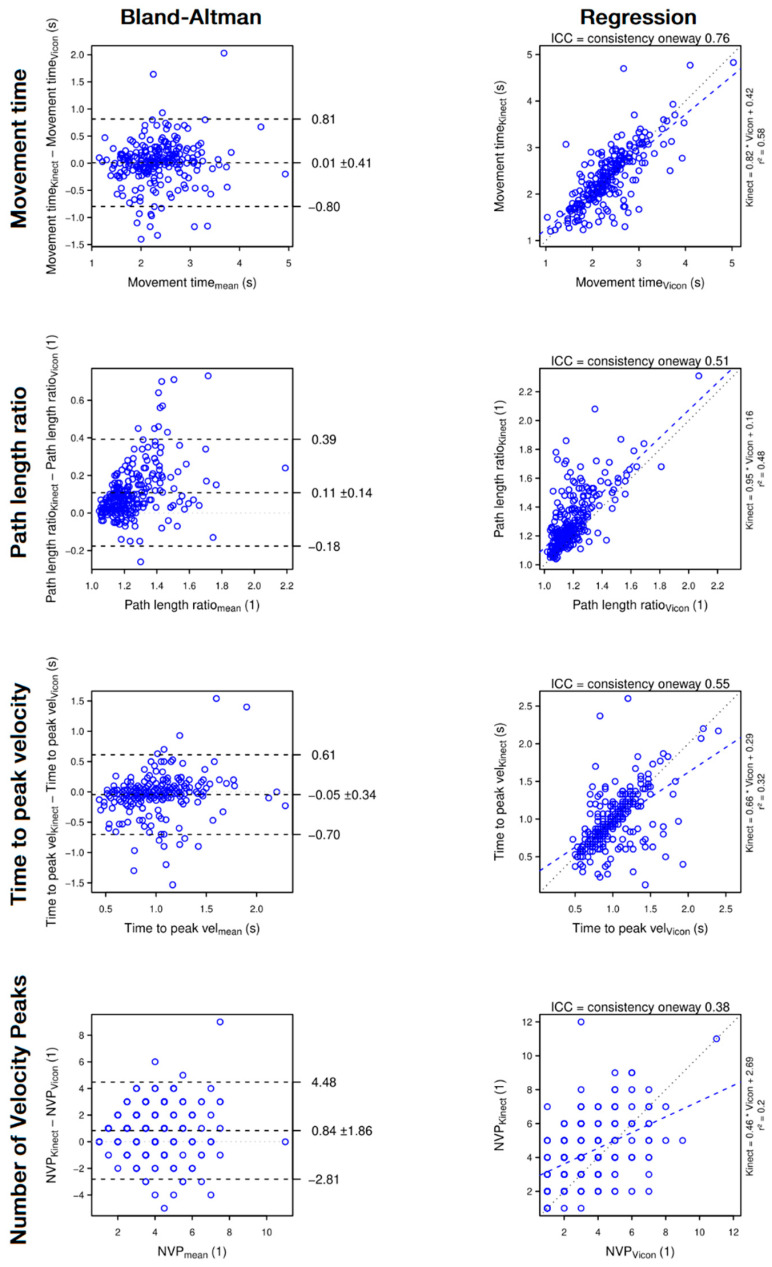
Bland–Altman plots (**left panels**) and regression plots (**right panels**) of movement time (1st row), path length ratio (2nd row), time to peak velocity (3rd row) and hand number of velocity peaks (4th row). The Bland–Altman plots show that the Kinect has almost no systematic error on the movement time and time to peak velocity but slightly overestimates the path length ratio and the number of velocity peaks. The regression plots show that, when assessed with the Kinect, the movement time is reliable, the path length ratio and the time to peak velocity are moderately reliable, but the number of velocity peaks is poorly reliable.

Speed

**Figure 6 sensors-22-02735-f006:**
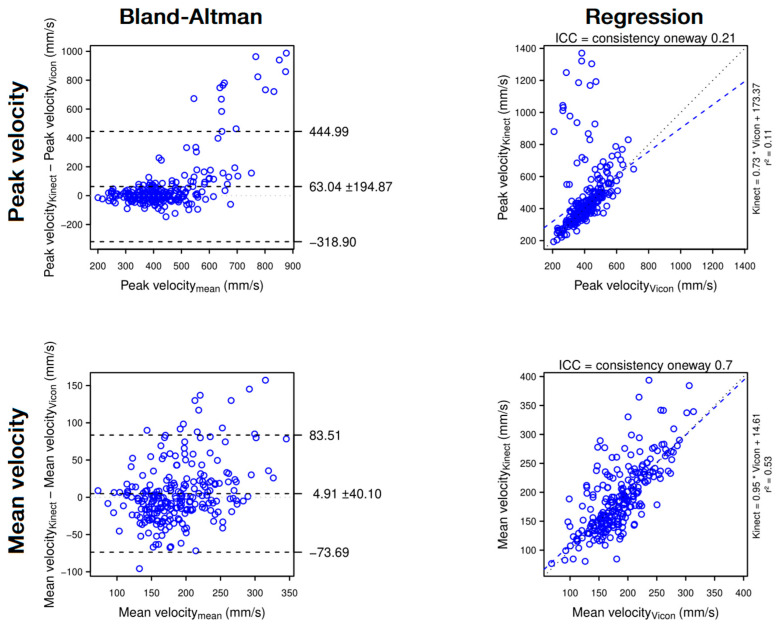
Bland–Altman plots (**left panels**) and regression plots (**right panels**) of peak hand velocity (1st row) and mean hand velocity (2nd row). The Bland–Altman plots show that the Kinect slightly overestimates peak hand velocity and mean hand velocity. The regression plots show that, when assessed with the Kinect, mean velocity is moderately reliable but peak velocity is only poorly reliable.

Displacements

The figure of displacements is in the Appendix A.

## 4. Discussion

Our study shows that in a horizontal reaching task, the Kinect measures trunk forward compensations with a good to excellent reliability and validity, but it is not sensitive to low amplitude trunk rotation. The Kinect also measures hand and trunk range of motion as well as movement time and mean hand velocity with a moderate to good reliability and with a good to excellent validity, respectively.

In contrast, the Kinect assesses variables involving elbow extension, shoulder flexion and shoulder abduction with a poor to moderate reliability and overall overestimates the variables. Finally, instantaneous Cartesian and angular measures with the Kinect are not precise enough, which artificially creates jerky movements and overestimates NVP, Path Length Ratio and Peak Velocity. Time to Peak Velocity is also affected resulting in a moderate reliability. The main results are summarised in Figure 7.

These data indicate that, although the Kinect is not reliable enough to analyse fine kinematics over time, the Kinect does allow for global motion analysis (such as range of motion, movement time and mean velocity).

### 4.1. Validation of the Kinematic Assessment Obtained by Kinect

The aim of this study was to assess the reliability and validity of the Kinect in measuring key kinematic variables used in the analysis of reaching after a stroke [41]. We did not record individuals with stroke in the present experiment, but we induced stroke-like movements exhibiting many key characteristics of reaching kinematics after a stroke, such as reduced movement speed, increased path length ratio, reduced shoulder abduction and increased trunk involvement [31].

#### 4.1.1. Trunk Motion

The Kinect has excellent reliability on trunk displacement (ICC = 0.93) and good reliability on shoulder displacement (ICC = 0.88) and anterior flexion of the trunk (ICC = 0.82). Because of a low mean trunk flexion (3.43°), the difference between means of −1.22 ± 3.05° overrepresent the *error_relative_* (−35%) in Figure 7.

In contrast, trunk rotation was poorly assessed with the Kinect (ICC = 0.38), in particular with an underestimation of up to 20° of low trunk rotation (Figure 4, top panel, left column) and other authors found the same result [20]. Note that the Vicon might also overestimate the low values of trunk rotation due to soft tissue artefacts [42]. Indeed, the trunk rotation being assessed mainly with the displacement of the shoulder marker on a transversal plane, a slight forward shift of the shoulder marker due to a shoulder flexion might artificially count for a trunk rotation. In any case, we propose that trunk rotation assessed with the Kinect should not be interpreted unless it exceeds 20°.

#### 4.1.2. Hand Motion

At the beginning of the reach, the Kinect sometimes confuses the forearm with the armrest. The Kinect correctly relocates the hand during the movement, creating a jerky correction. To a lesser extent, this temporo-spatial uncertainty occurs several times during the reach which artificially creates a jerkier movement and overestimates the NVP and the path length ratio (Figure 5, 2nd and 4th panel). By suddenly modifying the hand position, the Kinect creates a local velocity peak, possibly resulting in an overestimation of the peak hand velocity (Figure 6, top panel). Due to its relation to peak velocity, the time to peak velocity is also affected in some cases (Figure 5, 3rd panel). By averaging the velocity over the entire movement, the mean velocity resists these local uncertainties (Figure 6, bottom panel).

Because the reliability of Δ hand and Δ trunk assessment are good to excellent, variables derived from the Δ hand and Δ trunk measures such as PAU and PANU also have an excellent reliability, which confirms our previous findings [26].

#### 4.1.3. The Problem of Elbow in a Seated Reaching Task

First, the elbow is often confused with the back part of the armrest, resulting in a backward shift of the actual elbow position at the beginning of the reach. Second, due to the position of the hand located between the Kinect and the elbow, elbow occlusion can occur, leading to an error in the elbow position. Third, the elbow is poorly located on a fully extended arm (such as at the end of the reach) because of the alignment forearm—upper arm. For these reasons, Δ elbow and side variables such as elbow extension, shoulder flexion and shoulder abduction are only poorly to moderately reliable and should be interpreted with caution.

To reduce elbow occlusion, some authors suggest installing the Kinect higher [20], in front of the participant [35,36] and to perform functional movements in sitting [20,43], but the present study shows that this is not enough to achieve sufficient precision for clinical interpretation of some variables. Other authors suggest installing the Kinect on the ipsilateral side of the movement with an angle of 30 to 45° [44], and moving the Kinect when assessing the other side, but this makes the experiment more complicated and therefore reduces the speed and ease of use characteristic of the Kinect.

### 4.2. Smoothing out KINECT Errors

The present study shows that the occlusion of the elbow and the confusion between the forearm and the armrest produce large errors, resulting in a jerky movement due to high frequency noise (Figure 2). A first solution might be to use a chair with small armrests to decrease confusion errors. Furthermore, a correct filtering should be applied to the raw data. In fact, the analysis of raw data showed that the frequency band 0–2.5 Hz contains 95% of the spectral density. Applying a dual-pass 2nd order Butterworth type filtering with a cutoff frequency of 2.5 Hz greatly improved Cartesian kinematics (Table 2), while it had no effect on other variables (ICC changes ≤ 0.03). The ideal cut-off frequency for individuals with stroke might be higher than 2.5 Hz since people with stroke have more segmented movements than healthy individuals [2,45]. Thus, a too low cut-off frequency might remove important information about the movement. However, our study shows that range of motion and averaged variables do not suffer from a lack of filtering and can still be interpreted without filtering.

Other authors explored solutions to reduce Kinect errors. A deep learning algorithm applied to time series data reduced Kinect errors on shoulder and elbow range of motion by 88.8 ± 12.4% [46]. Another approach is to fusion the data of several Kinects to minimise occlusions and optimise limb tracking [47,48,49]. Ryselis reported an overall increase in accuracy of 15.7%. A combination of a Kinect and several Inertial Measurement Units (IMUs) could also be used to reduce the upper limb position error by up to 20%, according to Jatesiktat et al. [50,51,52]. Finally, a device-independent approach is to incorporate body constraints (such as human skeleton length conservation and temporal constraints) to enhance the continuity of the estimated skeleton [53]. These solutions have the potential to consequently improve the accuracy of the Kinect while remaining affordable, even if the solutions require a high technical level and might lengthen the duration of patient preparation.

### 4.3. Kinematic Assessment of the Upper Limb in Clinical Routine for Personalised Rehabilitation Post-Stroke

#### 4.3.1. Markerless Motion Capture Advantages for Kinematic Assessment

Due to its ease of use and markerless characteristic, the Kinect allows therapists to perform a simple kinematic assessment of the upper limb in about 5 min [5]. The software development kit (SDK) of the Kinect being openly available, the development of a software that automatically cleans the data and computes the valuable kinematic variables is also possible. In addition, if the rehabilitation department does not already use markerless motion capture for virtual reality rehabilitation, the low cost of such devices ensures their accessibility.

#### 4.3.2. Kinematic Assessment to Better Understand the Level of Motor Recovery of the Patient

A seated reaching task with and without trunk restraint gives the therapist valuable information to better understand arm-trunk use [5], but the very same task, when recorded with the Kinect, opens the door to a more comprehensive kinematic assessment.

The movement time, the mean hand velocity, the path length ratio and the time to peak velocity reflect the spatial and temporal efficiency of the movement. A low mean hand velocity or a low time to peak velocity, when combined with a high path length ratio or a high NVP, reflect the increased importance of feedback and corrections during the reach and therefore signal a decreased efficiency of open-loop control [54,55]. Except for the NVP, the Kinect measures these kinematic outcomes with acceptable accuracy-reliability (Table 1), making the Kinect a valuable tool for monitoring changes in upper limb kinematics.

The elbow, shoulder and trunk range of motion describe the motor strategy and quantify the level of compensation used by the patient. Reduced elbow extension and shoulder flexion signal a deficit in upper limb movement that is often compensated by an increase of trunk flexion and trunk rotation, and a freeze of shoulder adduction [1,12]. The Kinect measures these kinematic outcomes with varied accuracy-reliability, making the Kinect a valuable tool for monitoring trunk flexion, but not for monitoring shoulder and elbow movements.

Quantifying the non-use of the shoulder-elbow joints with the PANU score [5] indicates what the patient can do but does not do spontaneously. If the patient compensates with the trunk but can also perform the movement without using the trunk (PANU score > 6.5%), then at least some of the compensation is not mandatory to succeed in the task. In contrast, if the patient has compensation and is unable to perform the movement without this compensation (PANU score < 6.5%), then compensation is mandatory to succeed in the task [5]. This distinction is important because a compensation that improves reach efficiency should not be considered the same as a compensation which does not improve reach efficiency. The Kinect measures the PANU score with excellent accuracy-reliability (Table 1), which confirms previous results on patients [26].

#### 4.3.3. Early and Regular Kinematic Assessment to Individualise Rehabilitation, Enhance Motivation and Improve Recovery

Due to limited resources in stroke units, identifying individuals who are the most likely to benefit from rehabilitation has long been an issue [56]. We now know that the remaining upper limb function is the most promising factor to predict the upper limb recovery [57]. Thus, because kinematic monitoring is more sensitive to changes over the course of rehabilitation than clinical scales [8,9], this could allow specific therapy to be planned based on specific kinematic values or specific changes in kinematics. For example, therapy focused on arm use makes sense for patients with high PANU scores but is certainly less suitable for patients with mandatory compensations. In the same way that kinematic data measured during robot-assisted therapy can help predict patient recovery [58], using valid kinematic variables measured with the Kinect has the potential to predict which patients are the most likely to benefit from a specific therapy, and thus optimise patient care.

Therapists can also provide kinematic feedback to patients and set goals to involve them in rehabilitation. Fine-tuned feedback enhances motivation and increases the level of acceptance of treatment by patients [59]. Thus, providing kinematic feedback to patients leads to better recovery [60,61].

### 4.4. Limitations of the Study

This study faces several limitations. The experiment was conducted on healthy volunteers whose movement characteristics were experimentally manipulated to approximate those of people who have suffered a stroke. Though it is reassuring that we here replicate the main results obtained with a Kinect and with patients [26], the Kinect–Vicon comparison with patients with stroke is a necessary additional step to go beyond the basic results presented here.

Moreover, to induce a stroke-like movement, we asked healthy participants to hold a dumbbell during the reach. The presence of the dumbbell in the hand could have affected the ability of the Kinect to correctly locate the hand, even though this effect is likely small (i.e., the endpoint accuracy is similar for the loaded and unloaded hand and no significant difference was detected in the hand velocity root-mean-square error (RMSE_hand_) between conditions). The dumbbell could also have induced a greater occlusion of the elbow, which might have affected the performance of the elbow tracking, as evidenced by an increased error in the RMSE_elbow_ of 40 ± 72% between the unload and load conditions. However, this increased error in the elbow tracking is compensated by a wider dispersion in the load condition, and thus no improvement in the validity and reliability of the variables shown in this study was found in the unload condition.

Finally, the results presented here are only relevant for the same type of task, which is a unilateral horizontal seated reaching task. Results might differ in another type of task, such as non-horizontal reaching, or finger to nose test [62]. Indeed, due to the uncertainty of the depth measurement with the Kinect [17], movements in the frontal plane that are less dependent on depth changes might show a better accuracy than movements with variable depth when measured with the Kinect.

Despite these limitations, the results presented in this study are an important step towards rigorous validation of the Kinect tool for clinical assessment of upper limb kinematics, providing insight into reliable and unreliable kinematic variables. These results should be confirmed in a post-stroke population with a forthcoming clinical trial (https://clinicaltrials.gov/ct2/show/NCT04747587, accessed on 30 March 2022).

### 4.5. Future Work

There is a need to replicate the validation in a population with movement disorders, whether for a Kinect or for any other markerless system with similar benefits. In addition, as Kinect-like systems are increasingly used in virtual reality rehabilitation to monitor the recovery process [15,63], replication should focus on a wide variety of movements such as those used in virtual reality rehabilitation. Indeed, the present study assessed the validity of kinematic variables in a horizontal seated reaching task, but virtual reality protocols cover a much wider range of variables and tasks.

However, although Kinect is still widely used in virtual reality rehabilitation, Microsoft has not sold the Kinect since 2017. Instead, with the emergence of machine learning for image processing, many low-cost 2D joint tracking options have come to light [64], which have led to 3D motion reconstruction modules based on multiple RGB cameras and joint triangulation as provided by OpenPose [65], VideoPose3D [66] or Learnable Triangulation [67]. Microsoft has also aligned itself with the release of the Kinect Azure in 2020, a new version of the Kinect that uses a deep neural network (DNN) method instead of the random forest method used by the Kinect v2 [68]. In order to anticipate their growing use, future work should assess the feasibility, validity and reliability of these systems in a clinical or home environment.

## 5. Conclusions

The aim of this study was to assess the validity and reliability of the Kinect v2 for quantifying upper body kinematics with application to monitoring upper limb function after stroke. As a first step towards this goal, we induced stroke-like kinematics in healthy volunteers to better understand the validity of the Kinect for some of the key features of reaching kinematics compared to state-of-the-art 3D motion capture.

Our results show that the Kinect quantifies reaching efficiency, compensation with the trunk and shoulder-elbow nonuse with sufficient reliability. Our results also show that the Kinect does not quantify the number of velocity peaks and the peak hand velocity with sufficient reliability for clinical monitoring. Furthermore, as the elbow is poorly tracked by the Kinect during seated reaching, elbow extension, shoulder flexion and shoulder abduction should be interpreted with caution.

Further studies are required to validate the use of Kinect v2 and other markerless mocap systems in home or clinical contexts. Indeed, the quantitative variables that are adequately monitored by markerless motion capture can effectively supplement the clinical scales used by therapists. In addition, a periodic assessment of the deficit can allow precise longitudinal follow-up of motor recovery, which could improve the evaluation of the rehabilitation modalities and help to optimise the therapeutic pathway of patients. A final advantage of using lightweight, markerless motion capture devices in clinical routine is that an accumulation of kinematic data could allow ambitious retrospective studies to be carried out in the long run.

## Figures and Tables

**Figure 1 sensors-22-02735-f001:**
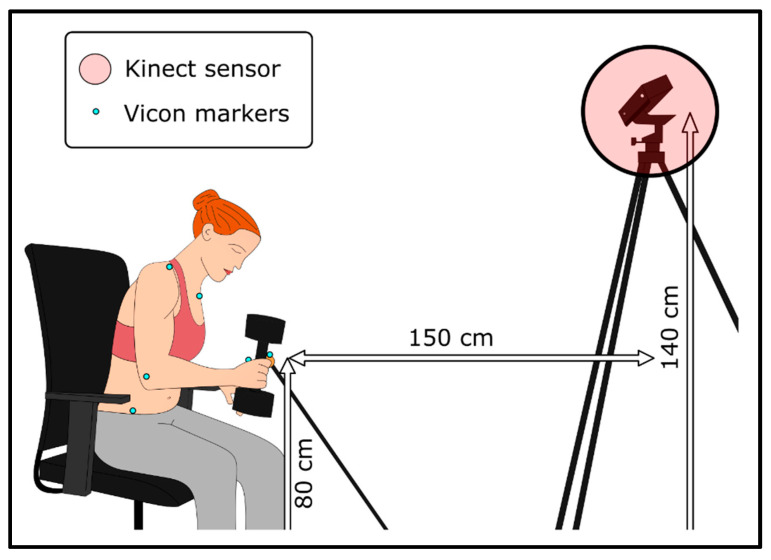
Experimental setup. 3D upper limb kinematics were simultaneously recorded by the Kinect and the Vicon motion capture systems. Vicon markers were placed on the target, hands, elbows, shoulders, manubrium and hips of the participant as close as possible to the joint centres located by the Kinect. The Kinect was located in front of the participant, at a distance of 1.50 m and a height of 1.40 m. The target (orange table tennis ball) was located at a height of 0.80 m, just within the anatomical reaching distance for the hand.

**Figure 2 sensors-22-02735-f002:**
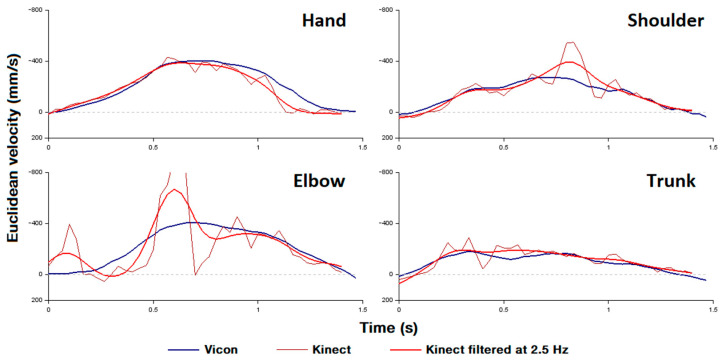
A selected example of the Euclidean velocity of a typical subject’s hand, shoulder, elbow, and trunk before and after filtering for the Kinect. The dashed line represents a constant Euclidean velocity of 0 mm·s^−1^. The figure shows how low-pass filtering reduces the instantaneous velocity errors of the Kinect compared to the Vicon measurements.

**Figure 7 sensors-22-02735-f007:**
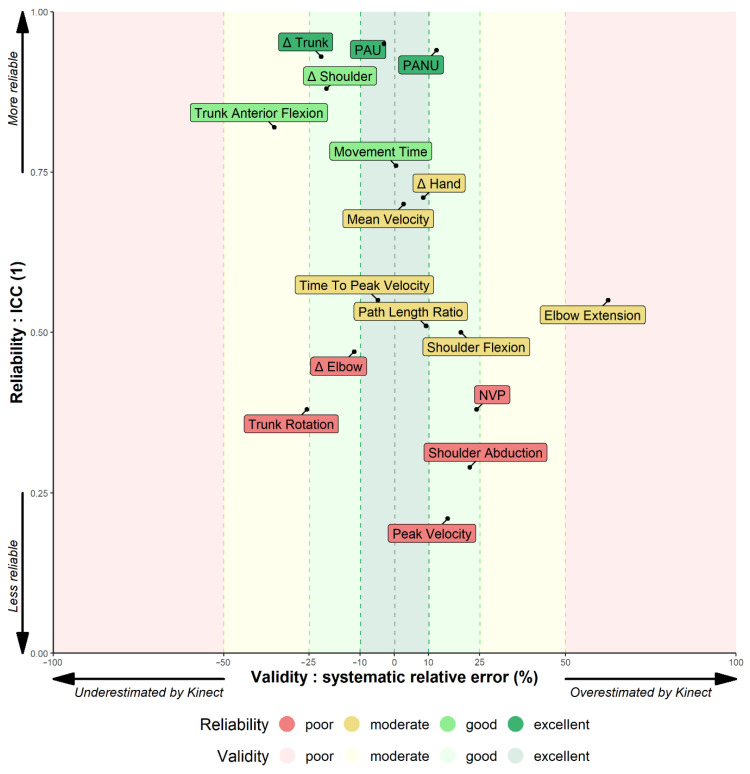
Summary of the validity and reliability of 17 kinematic variables assessed by the Kinect. The X axis represents the validity through the relative error. The Y axis represents the reliability through the one-way ICC. The closer the variable is to the centre on the X axis, the more valid it is. The higher the variable is on the Y axis, the more reliable it is. A perfect match between the Kinect and the Vicon values would place the variable at the centre on the X axis (error = 0) and at the top on the Y axis (ICC = 1). The figure shows that averaged postural and angular measurements are much more reliable than instantaneous Cartesian measures.

**Table 1 sensors-22-02735-t001:** Reliability and validity of the Kinect main kinematic variables used in the analysis of reaching in stroke. Reliability measures the consistency of the results compared to the ground truth (i.e., for each individual, how close the Kinect measure is to the ground truth). A perfect reliability between Kinect and Vicon data would result in an intraclass correlation coefficient (ICC) of 1, a coefficient of determination (r^2^) of 1, a Root-Mean Square Error (*RMSE*) of 0 and a Normalised Root-Mean Square Error (*NRMSE*) of 0 as well. Validity measures the extent to which the results are close to the ground truth on average (i.e., the higher the percentage of error on average, the lower the validity). A perfect validity would result in a difference in means of 0 in the Bland and Altman plot, and thus in a relative systematic error of 0.

Type	Variable	Reliability	ICC	r^2^	*RMSE*	*NRMSE*	Validity	Direction of Systematic Error	Absolute Systematic Error (Bias)	Relative Systematic Error	Figure
Angle	Elbow extension	**Moderate**	0.55	0.56	20.96°	24.59%	**Poor**	Overestimation	+13.95° (±15.67)	+62.5% (±70.2)	Figure 3—Top panel
	Shoulder abduction	**Poor**	0.29	0.13	11.72°	24.53%	**Good**	Overestimation	+2.67° (±11.44)	+22.0% (±94.7)	Figure 3—Middle panel
	Shoulder flexion	**Moderate**	0.50	0.37	13.13°	26.52%	**Good**	Overestimation	+6.61° (±11.37)	+19.4% (±33.4)	Figure 3—Bottom panel
	Trunk anterior flexion	**Good**	0.82	0.73	3.28°	11.40%	**Moderate**	Underestimation	−1.22° (±3.05)	−35.3% (±88.9)	Figure 4—Top panel
	Trunk rotation	**Poor**	0.38	0.25	9.17°	23.94%	**Moderate**	Underestimation	−4.10° (±8.22)	−25.7% (±51.6)	Figure 4—Bottom panel
Efficiency	Movement time	**Good**	0.76	0.58	0.41 s	10.26%	**Excellent**	Overestimation	+0.01 s (±0.41)	+0.4% (±17.6)	Figure 5—Top panel
	Path length ratio	**Moderate**	0.51	0.48	0.18	17.20%	**Excellent**	Overestimation	+0.11 (±0.14)	+9.2% (±11.7)	Figure 5—Middle top panel
Planning	Time to Peak velocity	**Moderate**	0.55	0.32	0.34 s	17.50%	**Excellent**	Underestimation	−0.05 s (±0.34)	−5.0% (±33.7)	Figure 5—Middle bottom panel
Smoothness	Number of Velocity peaks	**Poor**	0.38	0.20	2.04	20.36%	**Good**	Overestimation	+0.84 (±1.86)	+24.0% (±53.8)	Figure 5—Bottom panel
Speed	Peak velocity	**Poor**	0.21	0.11	204.42 mm·s^−1^	40.93%	**Good**	Overestimation	+63.04 mm·s^−1^ (±194.87)	+15.5% (±47.9)	Figure 6—Top panel
	Mean velocity	**Moderate**	0.70	0.53	40.32 mm·s^−1^	16.42%	**Excellent**	Overestimation	+4.91 mm·s^−1^ (±40.10)	+2.7% (±21.7)	Figure 6—Bottom panel
Displacements	PANU	**Excellent**	0.94	0.92	4.80	8.79%	**Good**	Overestimation	+1.88% (±4.43)	+12.3% (±28.7)	Appendix A— 1st panel
	PAU	**Excellent**	0.95	0.96	4.64	5.62%	**Excellent**	Underestimation	−2.88% (±3.65)	−3.1% (±4.0)	Appendix A— 2nd panel
	Δ Trunk	**Excellent**	0.93	0.95	19.68 mm	8.23%	**Good**	Underestimation	−15.04 mm (±12.72)	−21.5% (±18.2)	Appendix A— 3rd panel
	Δ Shoulder	**Good**	0.88	0.86	34.76 mm	11.30%	**Good**	Underestimation	−22.26 mm (±26.75)	−20.0% (±24.0)	Appendix A— 4th panel
	Δ Elbow	**Poor**	0.47	0.37	61.04 mm	17.14%	**Good**	Underestimation	−34.29 mm (±50.61)	−11.9% (±17.5)	Appendix A— 5th panel
	Δ Hand	**Moderate**	0.71	0.73	40.33 mm	14.54%	**Excellent**	Overestimation	+28.40 mm (±28.70)	+8.4% (±8.5)	Appendix A— 6th panel

**Table 2 sensors-22-02735-t002:** Reliability improvements through filtering. The filtering consisted of applying a 2nd order Butterworth filter with a cut-off frequency of 2.5 Hz on raw time series data. All variables shown in the table improved from filtering (ICC changes ≥ 0.12). Variables not shown in this table did not benefit from filtering (ICC changes ≤ 0.03).

	Movement Time	Path Length Ratio	Time to Peak Velocity	NVP	Peak Velocity
ICC before filtering	0.64	−0.12	0.19	−0.30	−0.05
ICC after filtering	0.76	0.51	0.55	0.38	0.21

## Data Availability

The code and the datasets generated and analysed during the current study are available in the Open Science Framework repository, https://osf.io/ckrdp/ (accessed on 30 March 2022).

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
