# Peer review of "Validity and Reliability of Kinect v2 for Quantifying Upper Body Kinematics during Seated Reaching"

_sensors, 2022, doi:10.3390/s22072735_

Round 1

Reviewer 1 Report

The paper with title “Validity and reliability of Kinect v2 for quantifying upper body kinematics during seated reaching” presents a novel contribution to the field but it needs some minimal corrections before being considered for publication: 1. could the root mean square error RMSE can be included in the table. For both the Kinect and Vicon? If possible, try to plot the difference of the results obtained with Vicon and Kinect. 2. It is suggested to the author to include a bode plot of the 2nd order Butterworth filter which evince the magnitude and phase of the filter in the frequency domain. Besides a fourier transform plot of the data is suggested. The reason is to verify and validate the data and designed filter proposed by the authors. 3. In section 4.2 please add a plot of the power spectrum density of the raw data that is pased through a low pass filter. 4. In sub-section 4.2 please specify why the cutoff frequencies of the 2nd order butterworth filter is selected. 5. As a suggestion to the authors, please in figures 2, 3, and 4 reduce the caption of the figures in order to be more understandable. 6. As a suggestion to the authors, please in figures 2, 3 and 4 add a legend to differentiate the kinect and vicon plots, so in this way the difference between the two tests can be verified easily.

Reviewer 2 Report

In this scientific paper, the main objective is to study validity and reliability of kinematic measures of upper body movements obtained by Kinect v2. The authors decided to test the specific measures acquired during a reaching task. This motor task is the most studied in the Neuroscience, in the Rehabilitation and in the Motor Control Framework. Therefore, there is a wide literature about reaching performance of several pathologic and healthy condition. Actually, in clinic, for evaluating upper limb functions there are many other motor tasks (the authors already report some of them such as nose-to-finger), but for the purpose of this study reaching seems reasonable. Moreover, the authors propose to simulate typical stroke motor performance asking to healthy people to hold a dumbbell while reaching the target. Some limitations could affect the results obtained but the authors sufficiently accounted about these limitations in the Discussion. Methods adopted for evaluate validity and reliability consist in comparing the same measures acquired with the Kinect v2 and the commercial (and largely used) motion capture VICON system. The algorithms used are standard, nothing innovative but they are appropriate. The results are in agreement with other studies and with common impression. However, the Kinect is an attractive low-cost and easy-to-use solution, and this type of study is useful for the scientific community in the field.

Reviewer 3 Report

This study aims to detect the reliability and validity of Kinect V2 for the assessment of upper limb reaching kinematics. The authors used the VICON as a gold standard for the analysis of the accuracy and reliability of the Kinect V2 in 26 healthy subjects during reaching tasks in a sitting position while emulating the altered motor pattern of individuals with stroke. The methods are well described and the English language is of good quality. However, there are two main issues in this manuscript that, in my idea, impede the scientific relevance of this work although well-conducted. 

First, I'm afraid there is lack of novelty in this study since the validity and reliability of Kinect V2 have been studied and verified enough in both healthy and stroke subjects (as you cited).

Second, healthy subjects can induce movements with a pattern similar to that of people with stroke but cannot imitate the overall condition of patients in a clinical setting. In addition, there are varying degrees of severity of motor impairment after stroke, and the movement strategies of stroke survivors differ at different stages of stroke (subacute or chronic). Normally, objective data are associated with clinical evaluations to address this aspect and to obtain reliable clinical conclusions. Therefore, it is methodologically and clinically inappropriate to draw conclusions about stroke survivors when a study was conducted exclusively in healthy subjects. 

This study could be considered as a feasibility study for the implementation of the next clinical trial on stroke subjects. Then, the introduction should be more focused on the altered upper body kinematics of stroke patients. Also, the outcomes should be interpreted carefully as influenced by the dumble.

The study presented in its current form, as well as the findings, don't reflect the study aim.

Minor comments:

Line 79-80. Please, move the info on the sample to the "Results". Add info on the participants' eligibility criteria (inclusion and exclusion).

Line 85. Add a brief description of the PANU test.

Line 94. What is exactly the light proprioceptive feedback? 

Line 96. The restrained trunk condition without a belt would NOT be expected from stroke people with upper limb and trunk impairment. So how can these results be comparable in the case of stroke? Please explain.

Did the authors take into account any specific clinical status that could be compared with their simulation? So it should be clarified in the methodology and considered in the interpretation of the data and in the discussion.

I would suggest reducing the figures and being more explicit in reporting the outcomes.

174-215, 223-239. No need for references in the title of the figures and tables. Prefer a concise title for the tables and figures.

Describe your findings in the main text "Results" rather than in the title of the tables/figures. 

Figure 6 and table 2 reported in the discussion are part of the study results.

Please, attenuate your interpretations when discussing the results regarding a clinical context or stroke survivors as this is not directly the case of your study.

Round 2

Reviewer 3 Report

The authors sufficiently revised the manuscript considering the previous comments. However, there are some minor concerns related to the manuscript that should be addressed before the final acceptance.

Line 26. "Stroke survivors have reduced range of motion in the upper limbs ...", the upper limb (without "s") sounds better in this context.
Line 29. This is not about the paretic "hand", especially regarding this study, but the affected/paretic side / upper limb / upper extremity ...
Line 108. Please report the extended form of the terms the first time in the text and then use the abbreviation throughout the manuscript: "Not using proximal arm" (PANU).
When discussing the value of the study please also add how the analysis of pre-treatment kinematic parameters can be important in the clinical application in terms of finding predictors of upper limb recovery and thus identifying individuals who may benefit more from specific treatments (Goffredo et al. "Kinematic data of the upper limb measured with retrospective robots from stroke patients are new biomarkers." Frontiers in neurology Vol. 12 803901. 21 December 2021, doi: 10.3389 / fneur.2021.803901) .
